# Location, Age, and Antibodies Predict Avian Influenza Virus Shedding in Ring-Billed and Franklin’s Gulls in Minnesota

**DOI:** 10.3390/ani14192781

**Published:** 2024-09-26

**Authors:** Matthew Michalska-Smith, Eva Clements, Elizabeth Rasmussen, Marie R. Culhane, Meggan E. Craft

**Affiliations:** 1Department of Ecology, Evolution and Behavior, University of Minnesota, Saint Paul, MN 55108, USA; michalsm@umn.edu (M.M.-S.); craft004@umn.edu (M.E.C.); 2Department of Plant Pathology, University of Minnesota, Saint Paul, MN 55108, USA; 3Department of Veterinary Population Medicine, University of Minnesota, Saint Paul, MN 55108, USA

**Keywords:** influenza A virus (IAV), gull, prevalence, transmission, *Larus delawarensis*, *Leucophaeus pipixcan*, Charadriiformes, positivity, seroprevalence, seropositivity

## Abstract

**Simple Summary:**

Each year, influenza infects millions of people, wildlife, and livestock, resulting in more than 30,000 deaths and a cost of more than USD 1 billion annually in the United States alone. In between the human flu seasons, the virus is maintained in wild bird populations, especially shorebirds and waterfowl. Much of the previous research has focused on ducks’ role in this disease cycle; however, there is a growing realization that gulls may provide critical links across space due to their long annual migrations. Yet, little is known of how influenza circulates in gulls. In this work, we evaluate the prevalence of influenza in two species of gulls in Minnesota, USA. We found important differences between species and gull age classes, highlighting the variability in disease spread through wildlife populations, with consequences for how and when the virus might be transmitted to humans/livestock. Importantly, we consider not only ongoing infections, but also look for antibodies that indicate past infections, and which may provide some level of future protection from disease. We find this additional level of surveillance to be beneficial in understanding the spread of disease, especially when researchers are trying to make the most of limited sampling.

**Abstract:**

Influenza A virus (IAV) is a multi-host pathogen maintained in water birds and capable of spillover into humans, wildlife, and livestock. Prior research has focused on dabbling ducks as a known IAV reservoir species, yet our understanding of influenza dynamics in other water birds, including gulls, is lacking. Here, we quantify morphological and environmental drivers of serological (antibody detection by ELISA) and virological (viral RNA detection by PCR) prevalence in two gull species: ring-billed (*Larus delawarensis*) and Franklin’s (*Leucophaeus pipixcan*) gulls. Across 12 months and 10 locations, we tested over 1500 gulls for influenza viral RNA, and additionally tested antibody levels in nearly 1000 of these. We find substantial virus prevalence and a large, nonoverlapping seroprevalence, with significant differences across age and species classifications. The body condition index had minimal explanatory power to predict (sero)positivity, and the effect of the surrounding environment was idiosyncratic. Our results hint at a nontrivial relationship between virus and seropositivity, highlighting serological surveillance as a valuable counterpoint to PCR. By providing indication of both past infections and susceptibility to future infections, serosurveillance can help inform the distribution of limited resources to maximize surveillance effectiveness for a disease of high human, wildlife, and livestock concern.

## 1. Introduction

Spillover of multihost pathogens between humans, domestic animals, and wildlife is increasingly common [1,2,3,4]. Pathogen spillover can have negative conservation implications for wildlife [5], cripple livestock farming economies and communities [6,7], and occasionally result in devastating global pandemics in humans (including SARS-CoV-2 from bats; Refs. [4,8]). In order to understand spillover risk and better control multihost pathogens, it is important to understand pathogen dynamics within the reservoir of infection [9,10]. Infection reservoirs can be defined as “one or more epidemiologically connected populations or environments in which a pathogen can be permanently maintained and from which infection is transmitted to the target population” [11], while the target population of interest can be humans or a specific animal species. Pathogen dynamics in infection reservoirs are driven by, among other things, host density, prevalence and intensity of infection, existing immunity, movement, and migration (which impact contact rates between infected and susceptible hosts; Refs. [9,12,13]).

Influenza A virus (IAV) is a classic multihost pathogen which is maintained in wild water birds, the natural reservoir host of the virus, and can spillover into humans (e.g., 1918 influenza pandemic; Ref. [14]), pigs [15], wild birds of conservation concern (e.g., the Caspian Tern *Hydroprogne caspia* and Common Tern *Sterna hirundo*; Ref. [16]), and domestic poultry (e.g., chickens and turkeys; Refs. [17,18]). Specifically, 16 of the 18 hemagglutinin subtypes of influenza A have been found in ducks (primarily mallards), gulls, or shorebirds [19], where IAV circulates year round with seasonal peaks of infection occurring when immature birds migrate en masse to and from the breeding grounds [20]. When migratory birds come in contact, either directly or indirectly, with domestic poultry, the likelihood of interspecies transmission increases with the number of wild birds and the frequency and duration of visits with domestic poultry [21]. In domestic poultry, IAVs are classified as highly pathogenic avian influenza (HPAI) or having low pathogenicity (LP or LPAI) based on laboratory criteria that include the presence of basic amino acids at the IAV hemagglutinin protein cleavage site and whether they cause mortality in at least 75% of intravenously inoculated susceptible chickens within 10 days post-inoculation [19]. Regardless of the IAV being classified as LPAI or HPAI, all IAVs have the potential to mutate and change and are an ongoing threat for susceptible humans, pigs, wild birds, and domestic poultry [22,23].

Although known infection reservoir species such as dabbling ducks can survive most IAV infections (depending on the strain) and potentially distribute IAV, there are gaps in our understanding of avian influenza dynamics in birds such as gulls. Gulls are a neglected infection reservoir for IAV [24], that are not only commonly infected with IAV, but are a species that, when infected, facilitate IAV reassortment and genetic change [25]. As such, gulls could be important for IAV spillover by being a source population for IAV from wild birds to humans [26] or from wild birds to domestic birds [27]. Furthermore, gulls also migrate long distances (i.e., internationally) and could bring novel influenza strains to new areas [28,29]. Since the literature review conducted by [24], IAV surveillance in gull species has increased and has been conducted more globally. However, the majority of the surveillance has occurred in Europe [25,30,31] and the studies in North America have primarily focused on the Atlantic flyway and Arctic regions [26,32,33,34,35,36,37], with studies of inland gulls less commonly conducted [29]. Regardless of their location and relative paucity, studies like these have demonstrated that gull IAV infection dynamics are quite variable due to differences in species, geography, diet, local ecology, and focal ecosystem.

To further add to the body of knowledge regarding gulls as an IAV infection reservoir, we studied IAV dynamics in two species in the state of Minnesota, U.S.A.: ring-billed gulls (*Larus delawarensis*) and Franklin’s gulls (*Leucophaeus pipixcan*). Poultry and grain farmers frequently report large flocks of gulls on farms and in fields and have questioned the role these birds may play in distributing avian influenza on the landscape. In the 2021/2022 HPAI H5 outbreak in North America, spatiotemporal analyses of HPAI-positive premises in the United States showed that HPAI detection was most likely to occur within seven days of heavy wild bird observations [38]. Seasonally, gulls, like all wild birds, have variable locations and activities in Minnesota, some of which bring them close to domestic poultry and others that preclude interactions with any domestic birds. For example, in the spring (February, March, and April), migratory gulls come back to their breeding grounds in Minnesota and are often sighted at landfills. In the summer, they are often found at a breeding colony where adults nest during the first part of the summer (May to early June). During the second part of the summer (late June through July), the breeding colonies have susceptible flightless hatch-year birds (with waning egg yolk-derived antibodies to IAV) that mix with adults. These immunologically naïve hatch-year birds are important for understanding IAV dynamics as they have higher IAV prevalence than the adults and those with a higher body condition index are less likely to be IAV positive [39]. In the fall (August, September, and October) the migratory gulls slowly leave Minnesota, during which time they are foraging at places like landfills to stock up for the migration and are mixing with other species.

The age of the bird, season of the year, and location are known drivers of IAV dynamics in wild birds, yet gulls have previously been studied at just the species level, rather than considering characteristics such as host age or other spatiotemporal characteristics. For example, three published studies in 2014–2016 all demonstrated a high seroprevalence and a low viral prevalence, findings which would discourage IAV surveillance in gulls since it appeared unlikely that IAV would be detectable in their seemingly immune populations [30,32,33]. However, surveillance and research efforts conducted on both inland and coastal gulls of Massachusetts revealed higher viral prevalence as detected by IAV PCR in juveniles [26], and further explored the relationship between seroprevalence, viral prevalence, and age in their multi-year longitudinal study. Ineson et al. [26] recommend “...targeted sampling of the sites where juvenile gulls first congregate on the mainland after leaving the colony, as this is where most virus will be detected”.

Encouraged by the efforts of IAV researchers who studied gulls of the coastal regions [26] and have experience with effective capturing methods and IAV assays [29,39], we conducted this research in an effort to explore drivers of serological (antibody detection by ELISA) prevalence and virological (viral RNA detection by PCR) prevalence in two species, ring-billed gulls (*Larus delawarensis*) and Franklin’s gulls (*Leucophaeus pipixcan*), in the non-coastal, i.e., inland/continental, state of Minnesota in the midwestern United States. We analyzed location effects to ascertain if gulls in Minnesota wetlands, i.e., wetlands historically postulated to harbor many wild IAV-positive *Anatidae*, were more likely to be IAV PCR positive than those in dense poultry production areas, i.e., areas with a highly dense population of IAV-susceptible domestic turkeys with previous HPAI and LPAI-infected flocks. In addition to location, we analyzed any correlations between age, weight, and species with antibody or viral RNA prevalence. Our goal was to further elucidate the drivers of IAV infections in gulls, thus generating information to help guide future surveillance efforts and control IAV interspecies transmission between wild and domestic animals, with implications for public health.

## 2. Materials and Methods

### 2.1. Gull Sampling

Using the methods described in Froberg et al. [39] and Rasmussen et al. [29], gulls were trapped, banded, measured, sampled, and released at landfills and breeding colony locations across Minnesota. Six of these locations were landfills that have generally maintained sizable numbers of gulls (>100) available to be captured and sampled. Three landfill sites were located within counties with a high density of poultry facilities (Blue Earth, Cottonwood, and Kandiyohi Counties) and three landfills were located in counties with a low density of poultry facilities (Dakota, Kanabec, and Rice Counties). Gulls were sampled between November 2016 and October of 2017.

When the birds were handled, the age was recorded as described in Froberg et al. [39]. Briefly, we recorded the age of birds as a categorical variable: hatch-year, juvenile, or adult. We based age on plumage, bill coloring, iris coloring, and leg coloring [40,41,42]. During the breeding season, birds incapable of sustained flight, i.e., in some stage of pre-juvenile molt, and judged to be approximately 3 to 5 weeks old, were categorized as hatch-year birds. This cohort is effectively considered immunologically naïve [43]. Based on phenotypic characteristics, we classified birds < 2 years of age as juveniles and birds > 2 years old as adults. In addition to age classification, the weight of each bird in grams (g) was measured using a spring scale (©PESOLA AG, Schindellegi, Switzerland). Finally, each bird’s keel (mm) and head-to-beak (mm) measurements were taken to calculate a body condition index (BCI), as outlined by Boersma and Ryder [44], namely the ratio of bird weight to the summed keel and head-to-beak lengths.

### 2.2. Influenza A Virus RNA and Antibody Detection

Briefly, adult and juvenile birds were bled from the brachial vein using a 23-gauge, one-half-inch-length needle and 3 mL syringe. The location of the basilic (also known as wing or ulnar) vein is similar in all avian species and is well described by Kelly and Alworth [45] for the domestic chicken and by Owen [46] for many wild avian species. The ring-billed gulls had 1.5 mL blood drawn, while Franklin’s gulls had 1 mL drawn as they are markedly smaller. All blood samples were placed in a 5 mL BD Vacutainer blood tube. After collection, blood samples were stored horizontally on ice in the field to allow for clot formation and serum separation. Within 24 h post-collection, the serum was poured off the clot into clean cryovials, then stored at −80 degrees Celsius until tested for AIV antibodies.

Birds were additionally swabbed (cloacal and oral) for virus RNA testing. To detect AIV in the swabs, the University of Minnesota Mid-Central Research and Outreach Center (Willmar, MN, USA) performed rRT-PCR via standard virus detection protocols according to Spackman and Suarez [47]. Briefly, viral RNA was extracted using a MagMAXTM-96 Viral RNA Isolation Kit (Applied Biosystems, Foster City, CA, USA) following the manufacturer’s instructions and using automatized robotic extraction equipment, the MagMAXTM Express-96 Deep Well Magnetic Particle Processor (Applied Biosystems). Real-time reverse-transcription polymerase chain reaction (rRT-PCR) was performed on the extracted RNA following the procedures, primers, and probe described by Spackman and Suarez [47] to detect the influenza A virus matrix gene. A specimen was considered positive for the AIV RNA if the cycle threshold (Ct) value was less than or equal to 39.5 [48]. Any bird swab with a CT value >39.5, or having no CT value, i.e., no amplification of viral RNA, was considered negative for IAV.

Sera were tested for IAV antibodies at MCROC with a commercial blocking enzyme-linked immunosorbent assay (bELISA, MultiS-Screen ELISA [FlockCheck], Idexx^™^, Westbrook, ME, USA). To determine seropositivity to IAV, a bird’s serum sample was considered negative for IAV antibodies if the sample result-to-negative control absorbance ratio (s/n ratio) was greater than or equal to 0.50. A sample was considered positive for IAV if the s/n ratio was less than 0.50 [49]. Sera with detectable antibodies by bELISA were forwarded to the Diagnostic Virology Laboratory at the USDA National Veterinary Services Laboratory in Ames, Iowa, USA for Hemagglutination Inhibition (HI) assays to detect IAV subtype-specific antibodies. The HI assays were performed via standard procedures according to Spackman and Suarez [47] and used the viruses listed in Table 1 as HI antigens. Continuous variables (i.e., CT value or s/n ratio) were compared between species/ages using two-sample Student’s *t*-tests, while discrete variables (i.e., “positive”/“negative”) were compared with 2-sample tests of equal proportions.

### 2.3. Generalized Linear Models

We sought to identify covariates that are most predictive of virus positivity and seropositivity using generalized linear regression models (GLMs) in which the response variable was binomially distributed (i.e., positive or negative test result). We identified the best-fitting model in a two-step process, first identifying the best way to incorporate location-specific effects (i.e., sampling site), and then considering the incorporation of other covariates.

For the first part, three potential methods for incorporating location effects in the model were considered. First, we included location as a fixed effect in the GLM, such that each location received a unique coefficient in the model. Second, we considered including location as a random effect; that is, where the location-specific coefficients were considered as independent, identically random draws from a distribution. Finally, we considered omitting explicit location from the models, instead including the percentage of habitat within a 10-mile radius of the sampling site that was classified as wetland (“wetland percentage”), and the number of commercially raised poultry within a 10-mile radius of the sampling site (“poultry production”), each of which were unique for each location. Wetland habitat classifications were provided by the National Wetland Inventory for Minnesota [50]. The number of poultry within a 10-mile radius was based on geospatial information obtained from Oklahoma State University’s Spatial Agrometrics Tool for Livestock Estimation (available at https://osu-geog.maps.arcgis.com/apps/MapSeries/index.html?appid=b1bf901f4e9b4ab7ae83798642b793b0, accessed on 25 May 2022) and in consultation with the Minnesota Turkey Growers Association and the Minnesota Board of Animal Health regarding proximity to poultry facilities. We then proceeded to use stepwise model selection on binomial GLMs that initially included location (included as identified above), species, age, body condition index, wetland percentage, and poultry production. Variables were algorithmically removed or re-added according to changes in Akaike information criterion (AIC) until the model could no longer be improved. From all models within two units of the lowest AIC value, the one with the highest area under the receiver operating characteristic curve was identified as the “best” model. Final model effects of categorical explanatory variables were explored in more detail using Tukey’s honestly significant difference test. Binomial model fit was further quantified using McKelvey–Zavoina Pseudo-R2 [51,52].

All analyses were conducted in R v. 4.4.1 [53], with functions from the broom [54], fuzzyjoin [55], GGally [56], ggbeeswarm [57], ggmap [58], ggpubr [59], GLMMadaptive [60], janitor [61], kableExtra [62], lme4 [63], lubridate [64], magrittr [65], multcomp [66], patchwork [67], pROC [68], scatterpie [69], and tidyverse [70] packages.

## 3. Results

### 3.1. Sampling Distribution

A total of 1592 gulls were sampled between October, 2016 and September, 2017 (Figure 1) at 10 locations across Minnesota, U.S.A., of which 244 were Franklin’s gulls (*Leucophaeus pipixcan*) and 1348 were ring-billed gulls (*Larus delawarensis*; Figure 2). Of these, 1588 were swabbed for influenza A virus RNA, and 990 of these had their blood drawn for antibody titer. The majority, 164 (67%), of Franklin’s gulls were adults, 79 (32%) were juvenile, and 1 was undetermined. For the ring-billed gulls, 726 (46%) were adults, 258 (16%) were juvenile, 297 (19%) were hatch-year, and 67 were undetermined. No hatch-year Franklin’s gulls were captured, as no sampling was conducted at Franklin’s gull breeding sites. IAV was detected in 424 of the swabbed birds (27% virus positivity), and 553 of the blood samples tested were found to positively detect IAV antibodies (54% seropositivity).

Analyses presented in the main text are restricted to site–date combinations in which at least five birds of each species were captured. This corresponds to birds captured at the Burnsville, Kandiyohi County, or Rice County landfills in August of 2017. Furthermore, we only considered samples for which both rRT-PCR and antibody titer results were available. Thus, 243 birds were considered in these analyses: 142 Franklin’s gulls (103 (73%) adults and 39 (27%) juveniles) and 101 ring-billed gulls (57 (56%) adults and 44 (44%) juveniles). IAV was detected in 145 of the swabbed birds (60% virus positivity), and 107 of the blood samples were found to positively detect IAV antibodies (44% seropositivity).

Correlations between all covariates are reported in Appendix A.

### 3.2. Observational Results

Body condition index (BCI) was found to have minimal explanatory power for virus positivity and seropositivity (Pseudo $R_{\text{McKelvey-Zavoina}}^{2}$ consistently less than 0.2), though, when significant, such relationships tended to be for virus positivity and positive in sign. That is, when such relationships existed, gulls with higher BCI were slightly more likely to be found with virus RNA. BCI did, however, differ significantly between species (t(133.9)=36.13, p<0.0001), and between ages within species (Franklin’s gulls: t(71.2)=3.663, p=0.0004; ring-billed gulls: t(78.683)=2.1975, p=0.0309; Figure 3).

Regardless of species, adult gulls were found to have higher seropositivity than juveniles ($\chi^{2}\left(1,N=243\right)=21.58$, p<0.0001), but not virus positivity ($\chi^{2}\left(1,N=243\right)=0.6722$, p=0.4123; Figure 4). Juvenile gulls were unlikely to be seropositive, while adult gulls were more likely to be seropositive than not. For virus positivity, both age classes were more likely to be found positive than not. For both virus- and seropositivity, Franklin’s gulls were found to be approximately equally likely to be positive or negative. Comparing species, ring-billed gulls were more likely than Franklin’s gulls to be found virus positive ($\chi^{2}\left(1,N=142\right)=12.33$, p=0.0004) and less likely to be found seropositive ($\chi^{2}\left(1,N=101\right)=11.57$, p=0.0007; Figure 5).

The effect of the surrounding habitat on virus/seropositivity was circumstantial; we found a slight positive relationship between Franklin’s gull virus positivity and the percentage of the surrounding habitat classified as wetlands (t(141)=4.28, p<0.0001, Pseudo $R_{\text{McKelvey-Zavoina}}^{2}=0.218$), and a slight negative relationship between ring-billed gull virus positivity and the estimated number of commercially raised poultry in the surrounding habitat (t(100)=3.79, p=0.0002, Pseudo $R_{\text{McKelvey-Zavoina}}^{2}=0.361$; Appendix A). Seropositivity was unaffected by either habitat variable (Appendix A).

When considering all available samples (i.e., rather than just the subset of sites/dates where both species were sampled), we find a significant negative relationship between virus positivity and seropositivity ($\chi^{2}\left(1,N=971\right)=57.07$, p<0.0001); however, this is not the case for the subsetted data ($\chi^{2}\left(1,N=249\right)=2.91$, p=0.0883). In particular, 57 of the sampled birds in the subsetted data were both virus positive and seropositive. Birds that were “dual” positive did not have significantly different PCR CT values or ELISA s/n ratios than did birds who were just virus- or seropositive. However, given seronegativity, virus-negative birds are slightly closer to our threshold for seropositivity (mean: 0.701, standard deviation: 0.143) than are virus-positive birds (mean: 0.779, standard deviation: 0.148; W(48,88)=1468, p=0.0034; Figure 6). There is no difference in S/N ratio (level of antibodies) of seropositive birds between virus-positive and virus-negative birds (W(50,57)=1538, p=0.4840). Thus, antibody titer does not predict virus detection in seropositive birds; however, if birds are seronegative, then virus positivity is correlated with higher SN ratios, i.e., weaker antibody response.

All analyses were likewise performed on the full dataset (Appendix A).

### 3.3. Model Results

The best-fitting model for virus positivity (for the subset of samples collected from dates, sites where at least five birds of each species were sampled) incorporated location, species, age, and body condition index, yet only the coefficients associated with location were found to be significantly different from 0 (using bold typeface to denote variables with significant model coefficients; PCRresult∼species+age+bci+location). In contrast, the best-fitting model for seropositivity included only species and age as explanatory variables (serologyresult∼species+age). Considering the quantitative response variables of cycle threshold (for PCR amplification) and s/n ratio (for ELISA), the best fitting models were PCRCT∼age+location and serologyS/N∼species+age, respectively.

The best-fitting model for virus positivity (for the full dataset) incorporated age and location as fixed effects, with both proving to be significant (using bold typeface to denote variables with significant model coefficients; PCRresult∼age+location). In contrast, the best-fitting model for seropositivity included species in addition to age and location, with all three significantly different from zero (serologyresult∼species+age+location). The best-fitting model for the PCR cycle threshold incorporates species, age, body condition index (BCI), and location, with only species not significant: PCRCT∼species+age+bci+location. For the ELISA result-to-negative control absorbance (s/n) ratio, the model included species, age, and location, all of which were significant: serologyS/N∼species+age+location.

## 4. Discussion

Influenza is a devastating, widespread disease in humans and livestock, but little is known of influenza dynamics in reservoir and “mixing vessel” species such as gulls. In this work, we identified differences in gull virus- and seroprevalence across space and time, but also by species and age. We found that younger gulls were unlikely to be seropositive, but more likely to be virus positive than not, while adult birds were often virus- or seropositive (or both). Between species, ring-billed gulls were more likely to be virus positive and seronegative, while Franklin’s gulls were equally likely to be positive and negative for both tests. Understanding these nuances of influenza disease dynamics can help inform surveillance and public and livestock health interventions.

### 4.1. Serological Sampling Can Complement Current Surveillance

Wildlife disease surveillance is often conducted through monitoring active infection status, for instance through the detection of pathogen RNA via PCR [71,72], and while such efforts provide valuable insight into the current disease dynamics, they are limited to providing a snapshot in time. In contrast, serology can reveal the signature of past infection [73], reducing the sensitivity of sample timing and providing insight into disease dynamics in other locations along migration routes despite temporally and geographically limited sampling.

For instance, in this work, sampling was restricted to four migration stopover sites in August of 2017. However, the lack of seropositive results by bELISA for juveniles of both ring-billed and Franklin’s gulls at that time allows inference that juvenile birds did not encounter IAV prior to their arrival at these sites [74] or that antibodies had waned to non-detectable levels. Furthermore, the relatively high positivity of bELISA and rRT-PCR results in adult gulls suggest that at least some of the adult gulls were infected with IAV prior to our sampling period. Our wider data include additional sampling at breeding sites and throughout the spring–early fall in which gulls are present in Minnesota (Figure 1 and Figure 2). These data show, for ring-billed gulls, higher seroprevalence in the spring (with nearly no virus positivity), followed by lower seroprevalence and much higher virus positivity in the fall. While in situ sampling of gulls across their migration would provide the most insight into real-time disease dynamics, sampling across the full spatial and temporal range of long-distance migratory species like gulls would require a substantial investment of time and resources.

Serosurveillance can also facilitate the forecasting of susceptibility to future outbreaks, because seropositivity as detected by bELISA and virus positivity as detected by rRT-PCR are negatively correlated. This negative correlation between seropositivity and virus positivity is supported by other studies of trans-hemispheric migratory Charadriiformes [75] and is in agreement with the widely accepted conclusion that age-related patterns of seropositivity and viral positivity observed in the wild result from birds gaining immunity to IAV with repeated infections over the course of their lifespan [76,77].

Moreover, serological surveillance needs not be restricted to a binary categorization. Our results revealed that seropositive gulls were more likely to be infected when their antibody titers were lower. However, seropositive birds had similar result-to-negative control absorbance ratio (s/n) ratios regardless of whether or not the virus was detected by PCR. Similarly, we find greater variance in antibody titers in general than rRT-PCR cycle thresholds, and while virus-positive gulls have statistically indistinguishable cycle thresholds across age or species (excepting lower cycle thresholds in hatch-year ring-billed gulls), serologically negative gulls’ s/n ratios varied across both age and species. Future work could explore these patterns more comprehensively, perhaps identifying key transitions in antibody titers that inform infection risk.

Our results suggest that if one wants to target surveillance to maximize the likelihood of positive test results, the virus is more detectable in juveniles than other age groups.However, if resources allow for it, the most information is gained by combining the serology and rRT-PCR results. Doing so for our data reveals that adults are actually slightly more likely to be positive for at least one test. While about 56% of adult gulls in our study were found to be virus positive, an additional 27% were seropositive and virus negative, with only 17% of the population actually having no known past infections.

Importantly, we did not find that seropositivity is a reliable indicator that a bird will be virus negative. Half of the seropositive birds in our study were also found to be virus positive (25% when considering all available samples), suggesting that either these birds have non-neutralizing antibodies and cannot fight off new infections (perhaps because these new infections are of a different strain than the previous infection) or that these birds are late in their infection timeline and are seroconverting (convalescing) and their antibodies are causing the virus to go down (but they temporarily have both virus and antibodies at detectable concentrations). If gulls are being reinfected while there is still the possibility of persisting virus from a prior infection, this raises the possibility of gulls serving as sources of recombination and new strain development [78], which would be a critical consideration for early detection of strains of interest for public health [79,80]. Future studies with higher resolution and repeated sampling of the same birds are needed to disambiguate these possibilities.

### 4.2. The Role of the Surrounding Environment

Interestingly, we found only idiosyncratic evidence of the surrounding environment (specifically the density of wetlands and poultry farms in the surrounding landscape) on prevalence. Franklin’s gulls were more likely to be positive as the proportion of wetlands increased, while ring-billed gulls were less likely to be positive as the number of poultry increased. The former could be driven by cross-species infections with other waterfowl in the surrounding wetlands, and can also be considered as an infection risk for those same populations [81]. The latter is harder to explain. Our expectation was that gulls are a potential infection risk for surrounding poultry agriculture, as gulls are often encountered in close proximity to poultry farms [82]. The negative correlation suggests that this risk might not be as large as suspected, as gull populations close to large farming operations are less likely to be carrying the virus.

Importantly, these relationships should be evaluated with caution, as there are only four sites considered in our main analysis. More extensive sampling in both space and time is needed to fully understand the relationship between the surrounding environment (natural and anthropogenic) on disease processes; however, such research is costly in terms of both money and labor.

Critically, some of these potential environmental covariates might likewise vary temporally; for instance, the actual coverage of wetlands might depend on recent precipitation. Our sampling revealed a significant variation in (sero)prevalence over time, with ring-billed gulls being largely virus negative (yet seropositive) in the spring/summer compared to the fall samples considered elsewhere in the text. This could be due to altering environmental conditions over time, or differential levels of pathogen exposure at other locations in the summer migration in between our sampling dates.

### 4.3. Insights into Influenza Reservoirs

Despite several decades of appreciation for the relevance of waterbirds for influenza A transmission dynamics, much of the current literature focuses on ducks as the focal reservoir for potential transmission to livestock and humans [24]. Our results point to the importance of broadening this perspective. Gulls, in particular, deserve increased attention, as they both migrate further than ducks [83] and are more commonly human associated [84]. Moreover, we find that influenza infections are common, with more than half of our samples being actively positive, and many more (especially adults) showing serological evidence of recent infection (Figure 4).

Gulls occur in relatively large numbers in the Midwestern U.S., yet are under-represented, compared to hunter-harvested ducks, in surveillance efforts in Minnesota, and arguably in the world [39]. Per data obtained from the Influenza Research Database [85], the surveillance for IAV in *Laridae* (gulls) and *Anatidae* (ducks) is quite disproportionately in favor of ducks (Table 2). Thus, despite the call by Arnal et al. [24] to expand surveillance to gulls and other *Laridae*, the gap in surveillance still remains.

Gulls migrate long distances [24,86], allowing for both long-distance disease transmission [87] and perhaps increasing recombination risk due to exposure to (and consequent infection with) a more diverse suite of strains. Upon returning to nesting grounds in the proximity of both humans and livestock, these gulls provide ample opportunities for introduction of new strains such as the recent outbreak of HPAI in North America (including Minnesota, USA) in 2021 [88].

Between the two gull species we considered in this study, our results suggest that while both species have relatively high prevalence, ring-billed gulls in particular showed widespread infection, suggesting they might be a more critical reservoir for IAV, especially those strains most adapted to gulls (e.g., H13/16). Critically, this relationship might differ for other, less-adapted viruses, including highly pathogenic H5 strains, which are more generalist and tend to result in high bird mortality.

It is yet unclear whether gulls in *Larus* and *Leucophaeus* genera have different susceptibility to, response to, and prevalence of influenza. To obtain information from the birds in their various life phases and locations, we performed cross-sectional sampling as the gulls moved through Minnesota during spring and fall migration and while raising chicks on their breeding colonies during the summer, and while we examined prevalence primarily, it would be interesting to explore susceptibility and response to AIV in the two species. Recently, Taylor et al. [89] described an outbreak in Herring Gulls (*Larus argentatus*) in Canada. Furthermore, the laughing gull (*Lecuophaeus atricilla*) has been studied in Europe and North America and is susceptible [33]. One might assume that all *Larus* sp. gulls are susceptible to influenza and respond similarly. However, there is limited evidence of this [90,91], and we agree with many avian researchers that gull susceptibility and response are an area of increasing interest [92]. Finally, of the reports of H5 HPAI infections in wild birds thus far in the United States in the 2022–2024 outbreak, there are 227 reports of gulls with H5 HPAI infections in the 2022–2024 outbreak, 12 of which are reports of ring-billed gulls, 0 of Franklin’s gulls, and 32 reports of gull-unidentified [93]. Whether this is a reflection of host susceptibility or host abundance requires further study.

### 4.4. Limitations of BCI in Cross-Species Analyses

Body condition indices (BCIs) are often developed as standardized measures of bird health [94]; however, our results call the applicability of this measure into question when comparing across ages or species. We found significant differences across species (with Franklin’s gulls having substantially lower values than did ring-billed gulls) and age (with juvenile birds having slightly lower BCI than adult birds in each species), though with substantial overlap in the distributions for the latter (Figure 3). These results suggest that BCI cannot be used reliably across birds of varying species or ages. More critically, while our previous results suggest that birds would have lower BCI during and immediately after infection [39], BCI was not found to relate strongly with the virus or seropositivity (best-fitting binomial glm $R_{\text{McKelvey-Zavoina}}^{2}=0.17$) and, when significant, was found to be positively associated with virus positivity. That is, birds with higher BCI were more likely to be found to be virus positive. Altogether, these results cast doubt on the usefulness of BCI in assessing disease status.

## 5. Conclusions

Despite being spatially and temporally constrained, this study provides insights into the distribution of influenza A virus in wild gull populations. Understanding the dynamics of this critical disease before being confronted by epidemics in livestock and human populations is critical to both preventing such outbreaks and adequately responding when they do occur. Our results highlight the heterogeneity in (sero)prevalence across age and species, as well as hint at the possibility of identifying environmental covariates of disease risk. In particular, the discrepancy of virus vs. seropositivity across ring-billed and Franklin’s gulls provides critical insights into differences in the ecology and epidemiology of avian influenza across species. The relationship between the ELISA result-to-negative control absorbance (s/n) ratio and the likelihood of virus positivity raises an important question for further analysis regarding the use of serosurveillance to predict future disease dynamics. Ultimately, these findings serve as a foundation for building hypotheses to direct further research, but can also be used to inform current surveillance, in particular emphasizing the benefits of adding serological sampling to future surveillance efforts and increasing attention to historically understudied gull species, which are nevertheless critical components in the spread of influenza through humans, livestock, and wildlife.

## Figures and Tables

**Figure 1 animals-14-02781-f001:**
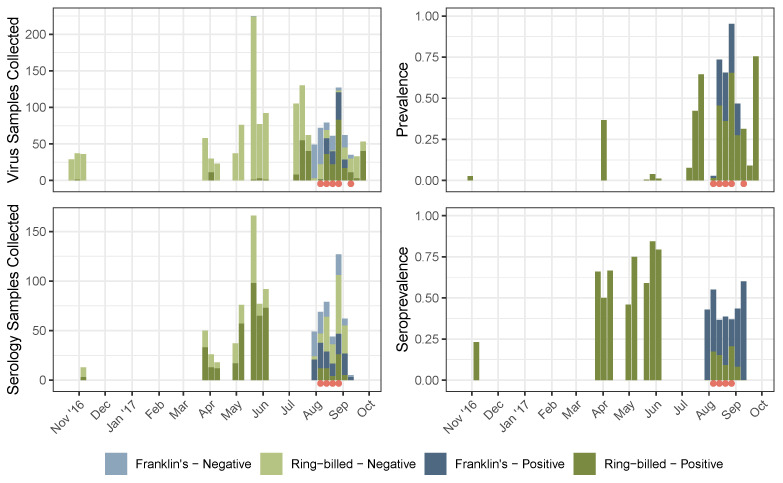
Timeseries of virus (**top**) and serology (**bottom**) sample counts (**left**) and positivity rates (**right**), aggregated to sampling week. Colors indicate species (blue for Franklin’s gulls, green for ring-billed gulls), while shade indicates test result (dark for positive, light for negative). Orange dots indicate week containing sampling dates included in the main text analysis, i.e., dates on which at least five birds of each species were sampled at a given site (N.b. not all samples in a given sampling week necessarily meet the criteria for inclusion; likewise, though virus samples were collected from birds of each species in the first week of September, different species were captured on different sampling dates/locations, thus precluding inclusion in our analysis.).

**Figure 2 animals-14-02781-f002:**
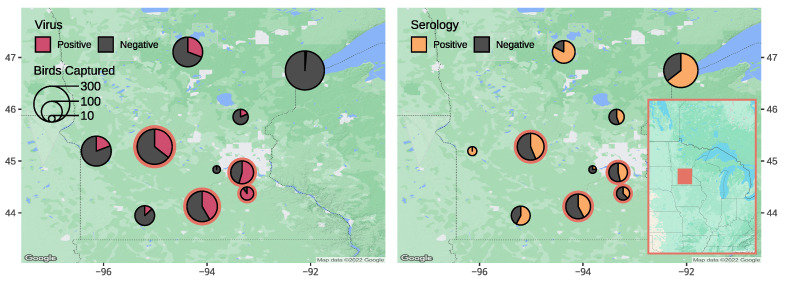
Map of the spatial distribution of sample counts and virus-/seropositivity rates at each site. Note that serology was not collected for all samples, resulting in different numbers of samples per site across the two panels. Pie-chart area scales with sample size are divided according to test result. These maps show all samples collected, with sites from which some sampling dates met our criteria for inclusion in the main text analysis (i.e., sites on which at least five birds of each species were sampled on a given date) are highlighted with orange outlines. The inset map shows our sampling region in a wider context. N.b. not all samples in an indicated sampling location necessarily meet the critieria for inclusion; only the subset of dates on which the criteria were met were included in the main text analysis.

**Figure 3 animals-14-02781-f003:**
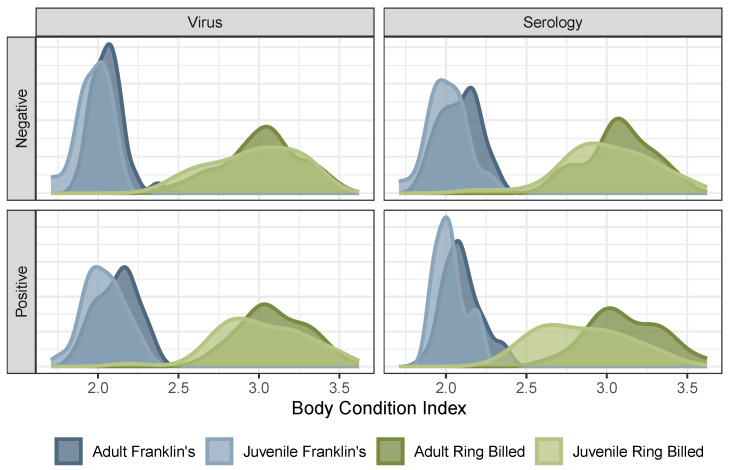
Body condition index (BCI) significantly differs between species (blue vs. green) and (to a lesser extent) age within species (light vs. dark within colors). This complicates the attribution of cross-species differences in IAV positivity to differences in BCI per se. Only includes samples collected from dates, sites where at least five birds of each species were sampled. See Appendix A for an analogous figure with the full dataset.

**Figure 4 animals-14-02781-f004:**
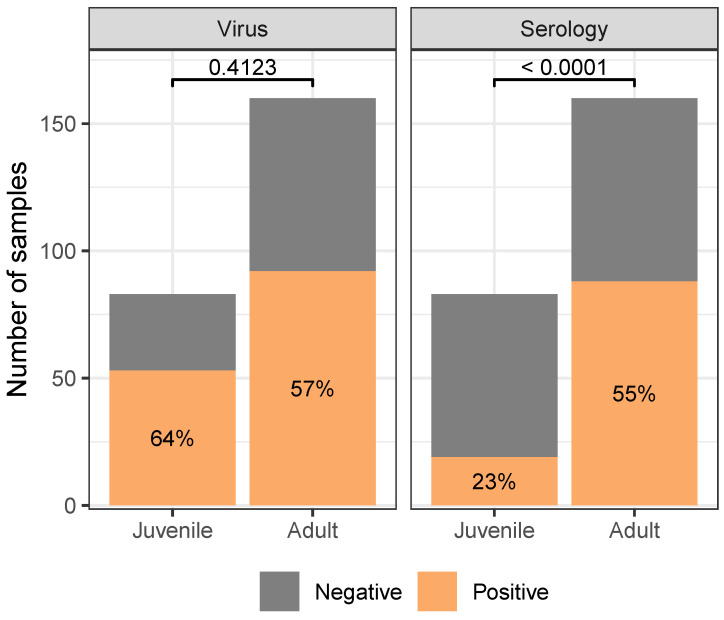
Gulls differed in their likelihood of being found seropositive, but not virus positive, between age classes (reported *p*-values are the results of 2-Sample Tests of Equal Proportions). Juvenile gulls were unlikely to be seropositive, while adult gulls were more likely to be seropositive than not. For virus positivity, both age classes were more likely to be found positive than not. Analogous figure considering continuous measures of test response (cycle threshold and result-to-negative control absorbance ratio in Appendix A). Only includes samples collected from dates, sites where at least five birds of each species were sampled. See Appendix A for analogous figures with the full dataset.

**Figure 5 animals-14-02781-f005:**
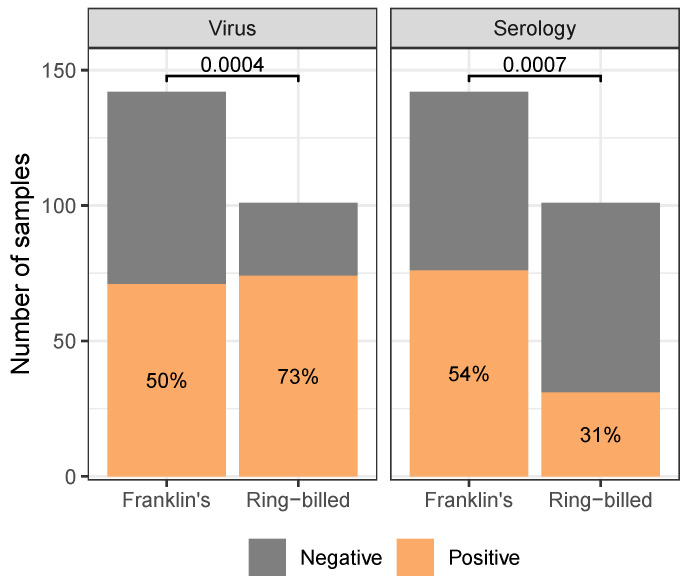
Gulls differed in their likelihood of being found virus- or seropositive between species (reported *p*-values are the results of 2-Sample Tests of Equal Proportions). While ring-billed gulls were more likely to be virus positive than Franklin’s gulls, they were less likely to be seropositive than Franklin’s gulls. For both virus- and seropositivity, Franklin’s gulls were found to be approximately equally likely to be positive or negative. Analogous figure considering continuous measures of test response (cycle threshold and result-to-negative control absorbance ratio Appendix A). Only includes samples collected from dates, sites where at least five birds of each species were sampled. See Appendix A for analogous figures with the full dataset.

**Figure 6 animals-14-02781-f006:**
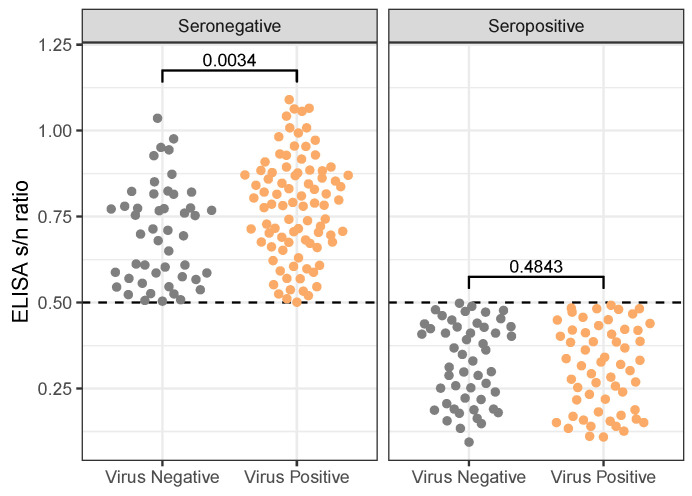
Relationship between ELISA result-to-negative control absorbance (s/n) ratio across different results for virus detection by PCR amplification (reported *p*-values are the results of 2-sample Wilcoxon Rank Sum Tests), and while gulls did not differ by virus detection in their s/n ratio when found to be seropositive, they did differ when seronegative, with virus-positive birds having slightly higher s/n ratios than virus-negative birds. That is, given seronegativity, virus-negative birds were slightly closer to our threshold for seropositivity (indicated by a dashed horizontal line) than were virus-positive birds. Interestingly, there is no difference in S/N ratio (level of antibodies) of seropositive birds that are virus positive and birds that are virus negative. Thus, antibody titer does not predict virus detection in seropositive birds; however, if birds are seronegative, then virus positivity is correlated with higher SN ratios, i.e., weaker antibody response. Only includes samples collected from dates, sites where at least five birds of each species were sampled. See Appendix A for an analogous figure with the full dataset.

**Table 1 animals-14-02781-t001:** Names and abbreviations of influenza A viruses used as antigens in the hemagglutination inhibition antibody assays. The abbreviation format is location/year/subtype/pathogenicity.

Virus Name	Subtype	Pathogenicity	Abbreviation	Lab Designator
A/Chicken/Mexico/1433-2/2008	H5N2	Low	MX/08/H5N2/LP	none
A/Turkey/Wisconsin/1968	H5N9	Low	WI/68/H5N9/LP	none
A/Turkey/Oregon/1971	H7N3	Low	OR/71/H7N3/LP	none
A/Turkey/Minnesota/9845-4/2015	H5N2	High	MN/15/H5N2/HP	133 ADV 1501

**Table 2 animals-14-02781-t002:** Summary of IAV surveillance results for *Laridae* and *Anatidae* species for eight years pre- and post-publication by Arnal et al. [24] using the cutoff date of 7 March 2013 from Arnal’s data inquiry.

	1/4/2005 to 7/3/2013			7/4/2013 to 12/31/2021		
	Positive	Tested	Positivity	Positive	Tested	Positivity
*Anatidae*	14,624	164,215	8.9%	29,584	258,631	11.4%
*Laridae*	1396	28,225	4.9%	3206	31,134	10.3%

## Data Availability

All data used in these analyses are available at https://doi.org/10.13020/9b7h-0973.

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
