# Peer review of "Location, Age, and Antibodies Predict Avian Influenza Virus Shedding in Ring-Billed and Franklin’s Gulls in Minnesota"

_animals, 2024, doi:10.3390/ani14192781_

Round 1
Reviewer 1 Report
Comments and Suggestions for Authors
- This study underscores the significance of avian influenza dynamics in two gull species, emphasizing the correlation between age, location, and seroprevalence. While the results are indeed intriguing, a few inquiries need to be addressed.
- Please add more details about Avian influenza reservoirs in the introduction.
- For extraction of viral RNA, please mention the method or kit used.
- I believe using a Ct value of 39.5 as a cutoff for PCR is quite high. Is this cutoff based on your lab's specific assay, or is it a standard value used generally?
- Is there any rationale for performing the HI assay with the three viral strains listed in Table 1?
- Since most of the samples were from Ring-billed Gulls, was this choice due to easier accessibility compared to Franklin’s Gulls?
- It would be valuable to explore the differences between Ring-billed Gulls and Franklin’s Gulls regarding virus positivity/negativity and seroprevalence. Consider discussing whether host responses (resistance or susceptibility to AI) might contribute to these differences.
- I find the conclusion to be somewhat general. It would be beneficial to highlight specific points from the results, which are quite rich and interesting.
Author Response
We thank the reviewer for their review and are pleased they found our work intriguing. We hope the reviewer agrees that the below responses/revisions have led to an improved manuscript.
Comment 1: Please add more details about Avian influenza reservoirs in the introduction.
Response 1: We now note that waterfowl/shorebirds are known carriers of nearly all influenza A HA subtypes (line 46-53):
"Influenza A Virus (IAV) is a classic multihost pathogen which is maintained in wild water birds, the natural reservoir host of the virus, and can spillover into humans (\eg 1918 influenza pandemic; Taubenberger & Morens 2020), pigs (Bourret 2018), wild birds of conservation concern (e.g. the Caspian Tern *Hydroprogne caspia* and Common Tern *Sterna hirundo*; Loeb 2022), and domestic poultry (e.g. chickens and turkeys; Garber, et al. 2016; Ssematimba, et al. 2019). Specifically, 16 of the 18 hemagglutinin subtypes of influenza A have been found in ducks (primarily mallards), gulls, or shorebirds (Spackman 2020), where IAV circulates year round with seasonal peaks of infection occurring when immature birds migrate en masse to and from the breeding grounds (van Dijk, et al. 2014)."
Comment 2: For extraction of viral RNA, please mention the method or kit used.
Response 2: We now note this information in the methods (line 160-169):
"To detect AIV in the swabs, the University of Minnesota Mid-Central Research and Outreach Center (Willmar, MN, USA) performed rRT-PCR according to standard virus detection protocols according to Spackman & Suarez (2008). Briefly, viral RNA was extracted using a MagMAXTM-96 Viral RNA Isolation Kit (Applied Biosystems, Foster City, CA, USA) following manufacturer's instructions and using automatized robotic extraction equipment, the MagMAXTM Express-96 Deep Well Magnetic Particle Processor (Applied Biosystems). Real-time reverse-transcription polymerase chain reaction (rRT-PCR) was performed on the extracted RNA following the procedures, primers, and probe described by Spackman & Suarez (2008) to detect the influenza A virus matrix gene."
Comment 3: I believe using a Ct value of 39.5 as a cutoff for PCR is quite high. Is this cutoff based on your lab's specific assay, or is it a standard value used generally?
Response 3: This is the standard used generally, especially in the State of Minnesota where a PCR cycle threshold (Ct) value of <40 is considered non-negative (https://www.bah.state.mn.us/hpai). For wild bird samples, Munster, et al. (2009) used a cutoff for negative samples of 40 after repeated evaluations of assay performance.
We have added the Munster reference to the text as well (line 170).
Comment 4: Is there any rationale for performing the HI assay with the three viral strains listed in Table 1?
Response 4: The antigens chosen were selected based on availability at NVSL in sufficient quantities, having positive control sera, reliability, repeatability, and representation of both H5 and H7 HPAI and LPAI strains that were circulating in the North America either more recently (2008 and 2014) or in the past (1968, 1971).
Comment 5: Since most of the samples were from Ring-billed Gulls, was this choice due to easier accessibility compared to Franklin’s Gulls?
Response 5: Yes, Franklin's Gulls are both less abundant in MN and more difficult to access and smaller than ring-billed gulls. Note, for instance, that there are 13,948 observations of Franklin's gulls in e-bird (https://ebird.org/species/fragul/US-MN), while there are 164,500 observations of Ring-billed gulls (https://ebird.org/species/ribgul/US-MN). This is due, in part, to the fact that, while both gull species are commonly observed in and adjacent to poultry farms in Minnesota, Ring-billed Gulls also nest on islands throughout Minnesota. Such locations can be difficult to access with surveillance materials (Pollet, et al. 2020), but with coordination, the islands are accessible and usually large enough, stable land bases on which teams can successfully work. Franklin’s Gulls, on the other hand, nest on floating platforms in marshy areas, making breeding colonies even harder to access (Burger & Gochfeld 2020). Therefore, we had access to Ring-billed gulls while in their breeding colonies AND at stopover locations (landfills), but Franklin's gulls were only accessible at the landfills.
Comment 6: It would be valuable to explore the differences between Ring-billed Gulls and Franklin’s Gulls regarding virus positivity/negativity and seroprevalence. Consider discussing whether host responses (resistance or susceptibility to AI) might contribute to these differences.
Response 6: We have added a paragraph to the discussion to this effect (line 425-440):
"It is yet unclear whether gulls in *Larus* and *Leucophaeus* genera have different susceptibility to, response to, and prevalence of influenza. To get information from the birds in their various life phases and locations, we performed cross-sectional sampling as the gulls moved through Minnesota during spring and fall migration and while raising chicks on their breeding colonies during the summer. While we examined prevalence primarily, it would be interesting to explore susceptibility and response to AIV in the two species. Recently Taylor, et al. (2023) described an outbreak in Herring Gulls (*Larus argentatus*) in Canada. Furthermore, the Laughing gull (*Lecuophaeus atricilla*) has been studied in Europe and North America and is susceptible (Guinn, et al. 2016). One might assume that all *Larus sp.* gulls are susceptible to influenza and respond similarly. However there is limited evidence of this (Ramis, et al. 2014; Brown, et al. 2006). In conclusion, we agree with many avian researchers that gull susceptiblity and response are an area of increasing interest (Verhagen, et al. 2021). Finally, of the reports of H5 HPAI infections in wildbirds thus far in the United States in the 2022-2024 outbreak, there are 227 reports of gulls with H5 HPAI infections in the 2022-2024 outbreak, 12 of which are reports of Ring-billed gulls, 0 of Franklin's gulls, and 32 reports for gull-unidentified (USDA-APHIS 2024). Whether this is a reflection of host susceptibility or host abundance requires further study."
Comment 7: I find the conclusion to be somewhat general. It would be beneficial to highlight specific points from the results, which are quite rich and interesting.
Response 7: We have added some explicit references to results we found interesting to our conclusion (line 462-467):
"In particular, the discrepancy of virus vs. seropositivity across Ring-billed and Franklin's gulls provides critical insights into differences in the ecology and epidemiology of avian influenza across species. The relationship between ELISA result-to-negative control absorbance (s/n) ratio and likelihood of virus positivity raises an important question for further analysis regarding the use of serosurveillance to predict future disease dynamics."
Reviewer 2 Report
Comments and Suggestions for Authors
In their article” Location, Age, and Antibodies Predict Avian Influenza Virus Shedding in Ring-billed and Franklin’s Gulls in Minnesota “the authors present a study about serological and virological prevalence of Influenza virus in gulls. Their aim is to elucidate the drivers of IAV infections in two species Ring-billed gulls (Larus delawarensis) and Franklin’s gulls (Leucophaeus pipixcan) in state of Minnesota. The goal of the study is to help with future surveillance and control Influenza A virus transmission between wild and domestic animals, with implications for public health.
The provided information is very interesting and the results can be of great help in the future surveillance and control of the disease.
I have only a few notes on it:
Introduction.
There are many repetitions of “gull”, almost in every sentence.
Materials and methods.
In my opinion, all figures are more proper for Section Results. In figures S1, S2, S3, S16, S17, S18 are shown some results. In subsection 2.1. you have to mention only materials, i.e. samples
Lines 139 and 153, it better to add the name of the author not only the number of the cite, for example standard methods according to Spackman
In subsection 2.3. there is information that is not for the materials and methods section (for example lines 177-180). Describe only the Models.
Results.
Figures S16-18 are more proper for subsection 2.1. I think that the comments under figures 2, 3 and 4 could be better to be in the main text, and also the text in Supplementary Material Section S5, is more proper to be included in subsection 3.3.
In the conclusion, the provided information is very useful and very important, because Influenza A viruses are still of a big concern and a potential danger for human and animal health.
Author Response
We thank the reviewer for their review and are pleased they thought our work would prove useful. We hope the reviewer agrees that the below responses/revisions have led to an improved manuscript.
Introduction.
Comment 1: There are many repetitions of “gull”, almost in every sentence.
Response 1: We have reduced our use of the word "gull".
Materials and methods.
Comment 2: In my opinion, all figures are more proper for Section Results. In figures S1, S2, S3, S16, S17, S18 are shown some results. In subsection 2.1. you have to mention only materials, i.e. samples
Response 2: We have moved figures S1 and S2 (now figures 2, 1) and the references to figures S16-18 to section 3.1. We have dropped figure S3.
Comment 3: Lines 139 and 153, it better to add the name of the author not only the number of the cite, for example standard methods according to Spackman
Response 3: fixed.
Comment 4: In subsection 2.3. there is information that is not for the materials and methods section (for example lines 177-180). Describe only the Models.
Response 4: We have removed this sentence as requested.
Results.
Comment 5: Figures S16-18 are more proper for subsection 2.1. I think that the comments under figures 2, 3 and 4 could be better to be in the main text, and also the text in Supplementary Material Section S5, is more proper to be included in subsection 3.3.
Response 5: As suggested, we have better incorporated the figure legends into the main text and moved Supplementary Material section S5 to subsection 3.3. We have, however, opted to keep the correlograms in the Supplementary Material to streamline the message of the main text. With respect to the inclusion of caption information in the main text, we have currently left the original captions as is, leading to duplicated information. If the reviewer or editor would like this duplication removed, we can simplify the captions as well.
Comment 6: In the conclusion, the provided information is very useful and very important, because Influenza A viruses are still of a big concern and a potential danger for human and animal health.
Response 6: We thank the reviewer for this positive note.
Reviewer 3 Report
Comments and Suggestions for Authors
This is a review for “Location, Age, and Antibodies Predict Avian Influenza Virus Shedding in Ring-billed and Franklin’s Gulls in Minnesota”
In this manuscript, the authors provide the results of a large sampling project in two species of gulls, comparing serology and virus isolation data to body weight, location, and age. The methods are explained well, and the authors provide an excellent orientation for context and importance. The only change that needs to be made is to switch the legends for Figures 2 and 3 so the correct graph is described.
Author Response
We thank the reviewer for their kind assessment of our work, and for catching the (embarrassingly) mismatched legends which have now been corrected.
Comment 1: The only change that needs to be made is to switch the legends for Figures 2 and 3 so the correct graph is described.
Response 1: fixed.
Reviewer 4 Report
Comments and Suggestions for Authors
To Editor, Authors
The manuscript “Location, Age, and Antibodies Predict Avian Influenza Virus Shedding in Ring-billed and Franklin’s Gulls in Minnesota”, Journal of Viruses is of interest for avian ecologists and virologists, especially for avian influenza researchers: for epidemiologists of avian influenza and other avian viruses during active surveillance.
I support this study! But it still has some questions to be addressed.
General comments:
1. It may be expected that younger gulls were unlikely to be seropositive, but more likely to be virus-positive than not, while adult birds were often virus- or seropositive (or both).
2. It is logic but the authors proved this with significant evidence with statistical original data - that is very big benefit of the article.
3. Generally, some figures presented in supplementary would be preferred in the main text.
4. I suggest to consider put Figure S1-S3to the main text, because it is very important for logic of the article.
5. In addition to the fact that you referred to articles using methods of catching and collecting samples from seagulls, it is extremely important to at least briefly describe such an important procedure as taking blood in vivo
6. Not so clear from the logic of the article I found how you chose the sample size. In some parts it is different. However I could miss something.
More specific:
7. Line 143: “Sera was separated from the blood” should add more details on this procedure
8. Line 270 – Standard testing approach is PCR instead of “genetic sequencing”
9. During serology studies sera with detectable antibodies by bELISA were forwarded for Hemagglutination Inhibition (HI) assays to detect IAV subtype-specific antibodies. As HI antigens you used H5 and H7 viruses. But I did not find the results about this issue. I suppose you could get no positives with such antigens.
Most probably the viruses circulated in gulls were H13 and H16 (gull-specific subtypes). I guess it would be nice to discuss in accordance with current Avian influenza virus ecology conception.
10. Lines 315-318 - “Half of seropositive birds were also found to be virus positive (25% when considering all available samples), suggesting that either these birds have non-neutralizing antibodies and cannot fight off new infections ……….” Again, here is the doubt about antibodies of different subtypes that could not have cross-reactivity between them.
11. Line 372. The role of this reservoir can be very high for seagulls of specific viruses such as H13 iN 16. But in this case, it is a completely separate reservoir, just for these viruses. Therefore, it should be emphasized that for other, less adapted viruses to seagulls, the picture may be different. In addition, the picture is completely different with respect to the highly pathogenic H5 virus, which usually causes the death of all birds, regardless of the taxonomic position, it is not specific to any particular bird species
Author Response
We thank the reviewer for taking the time to review our manuscript, and for their support. We hope the reviewer agrees that the below responses/revisions have led to an improved manuscript.
Comment 1: It may be expected that younger gulls were unlikely to be seropositive, but more likely to be virus-positive than not, while adult birds were often virus- or seropositive (or both).
Response 1: We agree with the reviewer that this result is not particularly surprising per se.
Comment 2: It is logic but the authors proved this with significant evidence with statistical original data - that is very big benefit of the article.
Response 2: We thank the reviewer for this positive note.
Comment 3: Generally, some figures presented in supplementary would be preferred in the main text.
Response 3: See response below.
Comment 4: I suggest to consider put Figure S1-S3 to the main text, because it is very important for logic of the article.
Response 4: We have added Figures S1 and S2 to the main text (now figures 2, 1), but have opted to drop Figure S3, as it replicates information in Figure S2 (now Figure 1). We have also modified the new figures to reflect the subsampling done for the main text. Note that the location of these figure references has been moved to results section 3.1 as requested by another reviewer.
Comment 5: In addition to the fact that you referred to articles using methods of catching and collecting samples from seagulls, it is extremely important to at least briefly describe such an important procedure as taking blood in vivo
Response 5: We have added detail to the methods (line 150-159):
"Briefly, adult and juvenile birds were bled from the brachial vein by using a 23-gauge, one-half inch length needle and 3ml syringe. The location of the basilic (aka wing or ulnar) vein is similar in all avian species and well-described by Kelly and Alworth (2013) for the domestic chicken and by Owen (2011) for many wild avian species. The Ring-billed gulls had 1.5mL blood drawn, while Franklin's gulls had 1mL drawn as they are markedly smaller. All blood samples were placed in a 5mL BD Vacutainer blood tube. After collection, blood samples were stored horizontally on ice in the field to allow for clot formation and serum separation. Within 24 hours post-collection, the serum was poured off the clot into clean cryovials, then stored at -80 degrees Celsius until tested for AIV antibodies."
Comment 6: Not so clear from the logic of the article I found how you chose the sample size. In some parts it is different. However I could miss something.
Response 6: We could interpret the reviewer's inquiry here in two ways:
First, the reviewer could be confused by our subsampling of the data for the main text analysis, while providing full dataset results in the supplementary material. If this is the case, we hope our newly modified figures 1 and 2 will aid in making this subsetting more clear.
Alternatively, the reviewer could be asking about the sampling of birds per se. If so, we would like to stress that the ultimate constraint was which birds happened to be caught at a given site on a given date, i.e. convenience sampling. Previous research has suggested that sample size is functionally determined by sampling effort and capture success rate (Froberg 2018, Froberg, et al. 2019).
More specific:
Comment 7: Line 143: “Sera was separated from the blood” should add more details on this procedure
Response 7: Please see response to comment 5.
Comment 8: Line 270 – Standard testing approach is PCR instead of “genetic sequencing”
Response 8: fixed.
Comment 9: During serology studies sera with detectable antibodies by bELISA were forwarded for Hemagglutination Inhibition (HI) assays to detect IAV subtype-specific antibodies. As HI antigens you used H5 and H7 viruses. But I did not find the results about this issue. I suppose you could get no positives with such antigens.
Most probably the viruses circulated in gulls were H13 and H16 (gull-specific subtypes). I guess it would be nice to discuss in accordance with current Avian influenza virus ecology conception.
Response 9: We agree the topic of particular strain identity is of great interest, however, this topic has already been discussed in some detail in a previous publication (Rasmussen, et al. 2023). As such, we do not feel it is appropriate to include in this work, which focuses more on the wider epidemiology of avian influenza in wild gulls. Nevertheless, we summarise some of the discussion from that other work here in case the reviewer would find it useful context:
Of the 144 Ring-billed gulls bELISA positive sera tested, only a single Ring-billed gull, MNAI0118, an adult bird from Blue Earth landfill, had a 1:16 HI titer against MN/15/H5N2/HP. Out of the 77 Franklin's gulls sera tested, two had HI antibodies detected: bird MNAI1175, an adult bird from Dakota County landfill, with a 1:16 HI titer against OR/71/H7N3/LP and bird MNAI1276, a juvenile bird from Rice County landfill, with a 1:32 HI titer against MN/15/H5N2/HP. All 218 other birds were negative for HI antibodies against the H5 and H7 LPAI and HPAI antigens used in the HI tests conducted at USDA NVSL DVL. Although high seroprevalence was detected by ELISA antibody tests, only rarely were H5 specific antibodies detected, which suggests that the majority of the birds were not exposed to H5 or H7 AIV nor did they possess detectable levels of cross-reactive antibodies by HI.
With respect to H13/16, as highlighted by the reviewer, gulls were found to have H13 per Rasmussen et al. (2023). In that work, it is noted that "Gulls are opportunists adapted to a range of urban and natural environments cohabitated by wild waterfowl, humans, and poultry (Ineson et al., 2022). Of the sixteen influenza HA subtypes (H1-H16) found in wild birds, two (H13 and H16) are found almost exclusively in gulls. H16 preferentially pairs with the N3 neuraminidase in gulls, while H13 pairs with N2, N6, and N8 (Postnikova et al., 2021). Despite the presence of predominantly H13 and H16 subtypes of AIV in gulls, gulls are susceptible to HPAI H5 infection (Ramis et al., 2014; Gulyaeva et al., 2016) and have not been spared from HPAI H5 outbreaks and mortality events; for example, 94 gull mortality/morbidity events have been recorded among the 5,552 detections of HPAI H5 in wild birds in the United States (USDA APHIS, 2023a). Recent studies document that gulls serve an important ecological role in the long-distance spatial diffusion of HPAI (Hill et al., 2022)."
Comment 10: Lines 315-318 - “Half of seropositive birds were also found to be virus positive (25% when considering all available samples), suggesting that either these birds have non-neutralizing antibodies and cannot fight off new infections ……….” Again, here is the doubt about antibodies of different subtypes that could not have cross-reactivity between them.
Response 10: We agree this was an oversight in our original submission. We now are more careful in our attribution of the immunizing benefits of detected antibodies throughout the draft.
Comment 11: Line 372. The role of this reservoir can be very high for seagulls of specific viruses such as H13 iN 16. But in this case, it is a completely separate reservoir, just for these viruses. Therefore, it should be emphasized that for other, less adapted viruses to seagulls, the picture may be different. In addition, the picture is completely different with respect to the highly pathogenic H5 virus, which usually causes the death of all birds, regardless of the taxonomic position, it is not specific to any particular bird species
Response 11: We thank the reviewer for highlighting this important caveat, we have modified this paragraph accordingly (lines 419-424):
"Between the two gull species we considered in this study, our results suggest that while both species have relatively high prevalence, Ring-billed Gulls in particular showed widespread infection suggesting they might be a more critical reservoir for IAV, especially those strains most adapted to gulls (e.g. H13/16). Critically, this relationship might differ for other, less adapted viruses, including highly pathogenic H5 strains, which are more generalist and tend to result in high bird mortality."